# Effect of Residents' Involvement with Small Hydropower Projects on Environmental Awareness

**Keigo Noda [1],\*** **, Kazuki Miyai [2], Kengo Ito [1] and Masateru Senge [1,3]**

[1]   Faculty of Applied Biological Sciences, Gifu University, 1-1 Yanagido, Gifu 501-1193, Japan; joroken@gifu-u.ac.jp (K.I.); senge@gifu-u.ac.jp (M.S.)

[2]   Graduate School of Applied Biological Sciences, Gifu University, 1-1 Yanagido, Gifu 501-1193, Japan; wolf.k7433@gmail.com

[3]   Union, Ltd., 1-1 Yanagido, Gifu 501-1193, Japan

\*   Correspondence: anod@gifu-u.ac.jp; Tel.: +81-58-293-2845

**Abstract:** Small hydropower plants utilizing unharnessed energy in existing irrigation systems are a prominent source of renewable energy. In Japan, land improvement districts play a key role in the management of irrigation systems, but face serious problems in terms of management sustainability and require participation from non-farmers. The purpose of this study was to examine the effect of residents' involvement in small hydropower projects on their environmental awareness and understanding of the projects' multifunctional regional value. We administered a questionnaire survey to 238 households in three areas: Itoshiro, Kashimo and Ibigawa. The respondents were categorized into four groups: participation, recognition, knowledge and control. Based on the degree of respondents' involvement in small hydropower projects, inclusive relationships between their involvement and awareness were revealed. These relationships suggest that the trigger of resident involvement is a key factor in developing sustainable small hydro facilities within existing irrigation systems.

**Keywords:** small hydropower; regional resource; participation; environmental awareness; irrigation system; land improvement district; participatory irrigation management; Long-Term Plan of Land Improvement; Japan

## 1. Introduction

Affordable clean energy is a key global challenge, and was raised in Sustainable Development Goal 7: Ensure access to affordable, reliable, sustainable, and modern energy for all, and we will increase substantially the share of renewable energy in the global energy mix by 2030 (SDG 7.2) [1]. Renewable energy is a fundamental requirement for sustainable development [2], along with drinking water [3] and education [4].

Renewable energy technologies, such as solar, wind, hydro and biomass, are dependent upon local resource availability [5,6]. A feasibility study of renewable energy in Australia, for example, examined only solar and wind because of the country's abundant sunlight and wind flows [7], while a global meta-analysis concluded that wind and small hydro are the most sustainable sources of electricity generation [8].

Small hydro is expected to be an alternative or supplemental energy source, especially in developing countries facing electricity shortages [9,10]. Hataya et al. [9] investigated small hydropower potential at eight sites in Delta-Egypt, and concluded that the estimated potential of 15.6 GWh would contribute to meeting the growing demand, and reduce dependence on fossil fuels. Uddin et al. [10] estimated the hydroelectric potential of Pakistan to be 60 GW, compared to the production of 7.2 GW

in 2016–2017, and concluded that the theoretical potential could satisfy current and future demands by 2050.

Small hydro in existing irrigation systems is attracting attention, particularly with regard to the synergies and trade-offs in relation to the water–energy–food nexus. Quaranta and Revelli [11] reviewed gravity water wheels, which are efficient and cost-effective micro hydropower converters. Adhau et al. [12] concluded that micro-hydropower generation in existing irrigation projects in India is both technically and financially feasible. Butera and Balestra [13] developed a methodology for characterizing an irrigation network and quantifying its hydroelectric production potential in the Piedmont Region of northern Italy. The use of pumps as turbines in existing irrigation systems, which utilize excess water pressure in the system, has been proposed as an integrated energy–water–environmental approach [14,15]. Perez-Sanchez et al. [16] reviewed energy recovery in existing water distribution networks, and concluded that it is a helpful alternative that can improve the energy efficiency of the system. While the above instances involve the additional installation of small hydro on existing irrigation systems, Ueda et al. [17] demonstrated a trade-off between irrigation and hydropower from dams in Japan involving seasonal demand changes among stakeholders.

Local networks with a common vision are needed in order to develop decentralized systems that can accommodate the distributed nature of renewable energy generation [18]. Participation in such local networks is essential for the activation of social capital and the establishment of vital communities [19]. Previous studies have examined willingness to participate in community energy projects. Kalkbrenner and Roosen [20] identified social norms, trust, environmental awareness and community identity as important factors in willingness to participate in community energy schemes; social norms play a larger role than environmental awareness. Bauwens [21] indicated that individual motivations to participate in community renewable energy projects are heterogeneous, and that levels of engagement are diverse. Although Yildiz [19] suggested that participation in a community affects both members and external stakeholders, there has been no clear investigation into the synergistic effects of participation in renewable energy projects and environmental awareness (e.g., volunteering in ecosystem investigations [22] or field trials studying the risk management of infection caused by irrigation with low-quality water [23]).

In Japan, the development of renewable energies is strongly promoted by the government, especially following the 2011 nuclear accident [24,25]. In 2018, fossil fuels, renewables and nuclear energy accounted for 77% (natural gas: 38%; coal: 32%; coal oil: 7%), 17% (hydro: 8%; solar: 6%; biomass: 2%; wind: 1%; geothermal: less than 1%) and 6% of total energy generation, respectively. The share of renewable energy contributing to electricity generation increased by 8% from 2010 to 2018, with the majority coming from solar and biomass (87% and 12% of the gains, respectively) [26]. Small hydropower plants that utilize unharnessed energy in existing irrigation systems are a prominent source of renewable energy. The Long-Term Plan of Land Improvement sets targets and projects for land improvement in 5-year intervals; the current Long-Term Plan of Land Improvement applies to the period 2016 to 2020 [27]. A policy target in the Long-Term Plan of Land Improvement is the promotion of the incorporation of small hydro into existing irrigation systems, with a goal of generating more than 30% of electricity consumption via irrigation systems.

The management bodies of small hydropower plants are diverse, and include prefectural and municipal governments, public companies and land improvement districts (LIDs) [28]. The LID is a type of farmers' organization, established by the Land Improvement Law for the purpose of the construction, improvement and management of irrigation/drainage facilities and land improvement projects [29]. LIDs play a key role in participatory irrigation management; Japanese irrigation systems are known to exhibit good participatory irrigation management practices [29,30]. Because LIDs face a serious management sustainability problem, due to farmer aging and resident diversification, the Long-Term Plan of Land Improvement promotes the participation of non-farmers in community activities (aspiring to their constituting up to 40% of the membership [27]).

Previous studies have revealed preferences or values that attract non-farmers to participate in participatory irrigation management. Hirose et al. [31] evaluated the preferences of non-farmers for water wheels used in irrigation canals in Okayama Prefecture, Japan. They extracted three factors that appeared to influence residents' preferences: regionality, functionality and environmental friendliness. Imai et al. [32] conducted a questionnaire survey with residents in Hyogo Prefecture, Japan, comprising 55% farmers and 45% non-farmers. They found that agricultural and the environmental values of irrigation ponds affected residents' behavior with respect to environmental conservation. They suggested that environmental values could increase the tendency to conserve irrigation ponds as natural resources. While those studies focused on the values associated with participation, Onimaru [33] analyzed the structure of non-farmer participation using a structural equation model, and found that participation in maintenance activities of irrigation and drainage canals was influenced by knowledge about irrigation and drainage water; in turn, that knowledge was influenced by experience.

Previous studies have documented factors affecting willingness to participate in community energy projects [20,21,31,32]. Kalkbrenner and Roosen [20], for example, found that social norms, trust, environmental awareness and community identity are important determinants. However, such studies have not focused on the synergistic effects of participation in renewable energy projects on environmental awareness, such as volunteering in ecosystem investigations [22], field trials in the risk management of infection caused by irrigation with low-quality water [23], or maintenance activities of irrigation and drainage canals [33]. In this study, we examined the effects of participation in small hydro projects on environmental awareness through a questionnaire survey in three areas: Itoshiro, Kashimo and Ibigawa in the central region of Japan.

## 2. Materials and Methods

### 2.1. Study Area

Three areas in Gifu Prefecture, Japan, with small hydropower plants were investigated. The parameters of the plants are listed in Table 1.

**Table 1.** Parameters of hydropower plants.

| Parameter | Itoshiro | Kashimo | Ibigawa | |
|---|---|---|---|---|
| Name of power plant | Itoshiro banba seiryu | Kashimo seiryu | Isai | Ito |
| Management body | Local agricultural cooperative (Residents initiative) | Municipal government | Land improvement district | |
| Turbine type | Pelton turbine | Francis turbine | Kaplan turbine | Spiral water wheel |
| Generation capacity (kW) | 125 | 220 | 70.8 | 22.0, 10.2 |
| Maximum water use (m$^3$/s) | 0.14 | 0.46 | 3.6 | 1.6, 0.9 |
| Effective height (m) | 104.5 | 61.6 | 3.0 | 2.0, 1.6 |
| Launch date | June 2016 | February 2014 | February 2015 | March 2015 |
| Capacity factor (%) | 55 (specifications) | 91.7 (2014), 79.4 (2015) | 74 | 80, 81 |
| Project expense (million JPY) | 230 | 338 | 220 | 110 |
| Project expense share | Prefecture: 55% Municipal: 20% Resident: 25% | Government: 50% Prefecture: 25% Municipal: 25% | Government: 67% Prefecture: 17% Municipal: 17% | |

1. Itoshiro area

Itoshiro is in the upstream basin of the Wasabiso River, a tributary of the Kuzuryu River. Aging and falling populations are serious problems; the total population was 252 in 2016, and has decreased by a quarter in this half-century, and more than half of the population are individuals over 65 years of age. A local non-profitable organization (NPO) established in 2003 has been leading regional promotion. Currently, four small hydropower plants are operating; we focused on the Itoshiro-Banba-Seiryu hydropower plant (Table 1), which is managed by the local agricultural cooperative in which almost all residents are involved.

2. Kashimo area

Kashimo is in Nakatsugawa City, which is upstream of the Sira River, a tributary of the Kiso River. Nakatsugawa City is eager to develop an eco-friendly city by promoting renewable energy. There are three small hydropower plants in the city. We focused on one plant in the Kashimo area, the Kashimo-Seiryu hydropower plant (Table 1); this plant was established by Gifu Prefecture and is managed by the municipal government of Nakatsugawa City. Daily maintenance is delegated to the irrigation association in the Ogo sub-area inside Kashimo. The population of the Kashimo area in 2015 was 2815; in the Ogo sub-area in 2015, the population was 465.

3. Ibigawa area

Two small hydropower plants, Isai and Ito (Table 1), in the town of Ibigawa, which covers the basin of the Ibi River; these are managed by the Seino-Yosui LID. The total population in Ibigawa town was 8075 in 2016. This area is the most populated of the three study areas in this investigation, and two-thirds of the residents are not farmers.

*2.2. Interviews to Establish Project Organization*

We conducted interviews on project organizations prior to undertaking the questionnaire survey. The interview was composed of four items: project purpose, prior explanation to residents, uses of income earned by selling electric power, and effects on irrigation.

In Itoshiro, we interviewed the head of the local agricultural cooperative and two representatives of the NPO on 23 August 2016. We conducted two interviews at the project in Kashimo: one with a representative of Gifu Prefecture, and the other with a representative from Nakatsugawa City, on 7 February and 22 February 2017. We also interviewed a representative of the LID in Ibigawa on 1 May 2017.

*2.3. Questionnaire Survey*

We conducted the questionnaire survey with a representative of each household in the three study areas. The questionnaire asked about respondents' demographic information, awareness of environmental issues, and concern with small hydropower plants; the order of the questions was designed to avoid any effects of questions about small hydropower plants on answers regarding environmental awareness.

The respondents' demographic characteristics included their sex, age, hometown, years in the area and farming activity. We measured their awareness of domestic energy, local environment and regional promotion by four-step ordered responses: (1) "interested and actively involved"; (2) "interested and occasionally active"; (3) "interested but not active"; and (4) "not interested". We asked about the respondents' concerns in terms of their general knowledge of small hydropower plants and their recognition of and participation in local projects. General knowledge was measured by three-step ordered responses: (1) "know about"; (2) "sounds familiar"; and (3) "do not know". Recognition of the project was measured by four-step ordered responses: (1) "involved in the project"; (2) "know about"; (3) "sounds familiar"; and (4) "do not know". The degree of participation in the project was measured

by five-step ordered responses: (1) "participate actively"; (2) "participate with some interest"; (3) "participate for social reasons"; (4) "do not participate but interested"; and (5) "not interested".

Because the options were on an ordinal scale, the Mann–Whitney U-test (a nonparametric analysis) was applied to determine the significance of differences between order statistics, with significance adjusted by Bonferroni correction for multiple comparisons [34]. The test was conducted using the "exactRanTests" package in R-3.6.0.

We asked the NPO to circulate the questionnaire to all households in the Itoshiro area (2 July 2017) and then to collect them (18 December 2017); the total number of households was 110. For the Kashimo area, we asked the Ogo irrigation association to circulate (25 July 2017) and collect (3 August 2017) the questionnaires from all households in the town; the total number of the households was 141. In both areas, questionnaires were completed by the head of household. We recruited 101 households by random sampling in the Ibigawa area, because it was broader and more populated than the other two areas, but the sample size was similar. We conducted the survey by daytime face-to-face interviews between 6 October and 4 November 2017.

## 3. Results

### 3.1. Interviews with Management Bodies

The results of interviews with management bodies are summarized in terms of project purpose, coverage of prior explanation, and use of income by selling electric power (Table 2). The main purpose of the project in Itoshiro was to support the town, and they valued the participation by all residents in regional promotion programs. The prior explanation of the project was distributed to all residents, except two households in the area, by the representatives for each town; they also delivered the minutes of the incorporators' meeting. The incorporators provided additional explanations until residents agreed. The new water allocation for power generation was accepted by residents because the representatives prioritized the original use, namely irrigation, and explained the plan in advance. The income earned by selling electric power was put toward the cost of streetlamps, community fees, and was used as a financial resource for revitalizing abandoned farmlands and operating community farming.

**Table 2.** Summary of the interview for the management body.

| Item | Itoshiro | Kashimo | Ibigawa |
|---|---|---|---|
| Project purpose | • Regional promotion | • Regional promotion<br>• Low carbon society | • Promotion of renewable energy<br>• Agriculture promotion |
| Coverage of prior explanation of the project | • Most residents | • Representative of each hamlet<br>• Irrigation community | • Representative of land improvement district (LID) |
| Use of income by selling electric power | • Streetlamps<br>• Re-cultivation of abandoned farmland | • Management of the plant<br>• Maintenance of the plant and land improvement project | • Allotted charge for irrigation<br>• Management and maintenance of the plant |

The small hydropower project in Kashimo had two major purposes; the first was its regional promotion via cost savings for land improvement and rural promotion facilities, and the second was its contribution to a low-carbon society by reducing $CO_2$ emissions. The Environmental Policy Division in Nakatsugawa city, which had jurisdiction over small hydropower projects, explained the project to the leaders of the towns in Kashimo, who acted as representatives for the residents. The municipal leaders also explained the new water intake for power generation to communities who used the

irrigation system, who easily accepted it because the low water reach was in the forest and none of the communities used that water for irrigation. The income earned by selling electric power was put toward the operating costs of the plant and land improvement facilities. Cleanup work at the plant inlet, as regular maintenance, was outsourced to the irrigation community; the cost was covered by income generated by the plant. The income was greater than expected, leading residents to support other plants in the Kashimo area.

The small hydropower project in Ibigawa was planned under a national promotion program for renewable energy. The Seno-Yosui LID, the management body for the plants, aimed to sustain agricultural production by allotting income from the sale of electric power to irrigation dues. The prior explanation covered the representative of the LID because the plants belonged to the LID and did not require input from non-farming residents.

*3.2. Results of the Questionnaire Survey*

3.2.1. Collection and Valid Response Rates

The numbers of samples, collections and valid responses are listed in Table 3. The collection rate in the Itoshiro area was the lowest (64%), and the collection rate in the Kashimo area was much higher (94%). However, response rates (=valid responses/distribution*100) in both areas were approximately 50%; 49% in Itoshiro and 61% in Kashimo. The collection and response rates in Ibigawa were almost 100%. The differences in the response rates were considered to result from differences in the survey methods. In Itoshiro and Kashimo, the survey was conducted indirectly, which probably led to the low collection rates and incomplete responses. In contrast, the survey in the Ibigawa area was conducted in a direct visit; answer sheets were collected by hand and respondents were supported when the questions were not clear.

**Table 3.** Distribution and valid responses for the questionnaire survey.

| Study Area | Distribution | Valid Responses (Rate) |
|------------|--------------|------------------------|
| Itoshiro | 110 | 54 (49%) |
| Kashimo | 141 | 86 (61%) |
| Ibigawa | 101 | 98 (97%) |
| Total | 352 | 238 (68%) |

3.2.2. Respondent Profiles

The respondents' profiles are summarized in Figure 1. More than 75% of the respondents were male in Itoshiro and Kashimo; 78% in Itoshiro and 85% in Kashimo. The male and female rates were equal only in the Ibigawa area. This difference was also likely due to the survey method. In the Itoshiro and Kashimo areas, the questionnaire was delivered to each household and completed by the head of the household, who, in Japan, is traditionally the oldest man in the household. In contrast, each household in the Ibigawa area was visited directly, and the respondent was not always the head of the household.

Half of the respondents in the Itoshiro area were in their 60s and 70s, and the youngest was in his 20s. In Kashimo, the majority of respondents were in their 60s (44%), followed by people in their 50s (26%); the proportion of respondents in their 70s was smaller. The youngest respondents were three people in their 30s. In Ibigawa, people in their 60s and 70s comprised two-thirds of respondents, and the youngest were four people in their 20s.

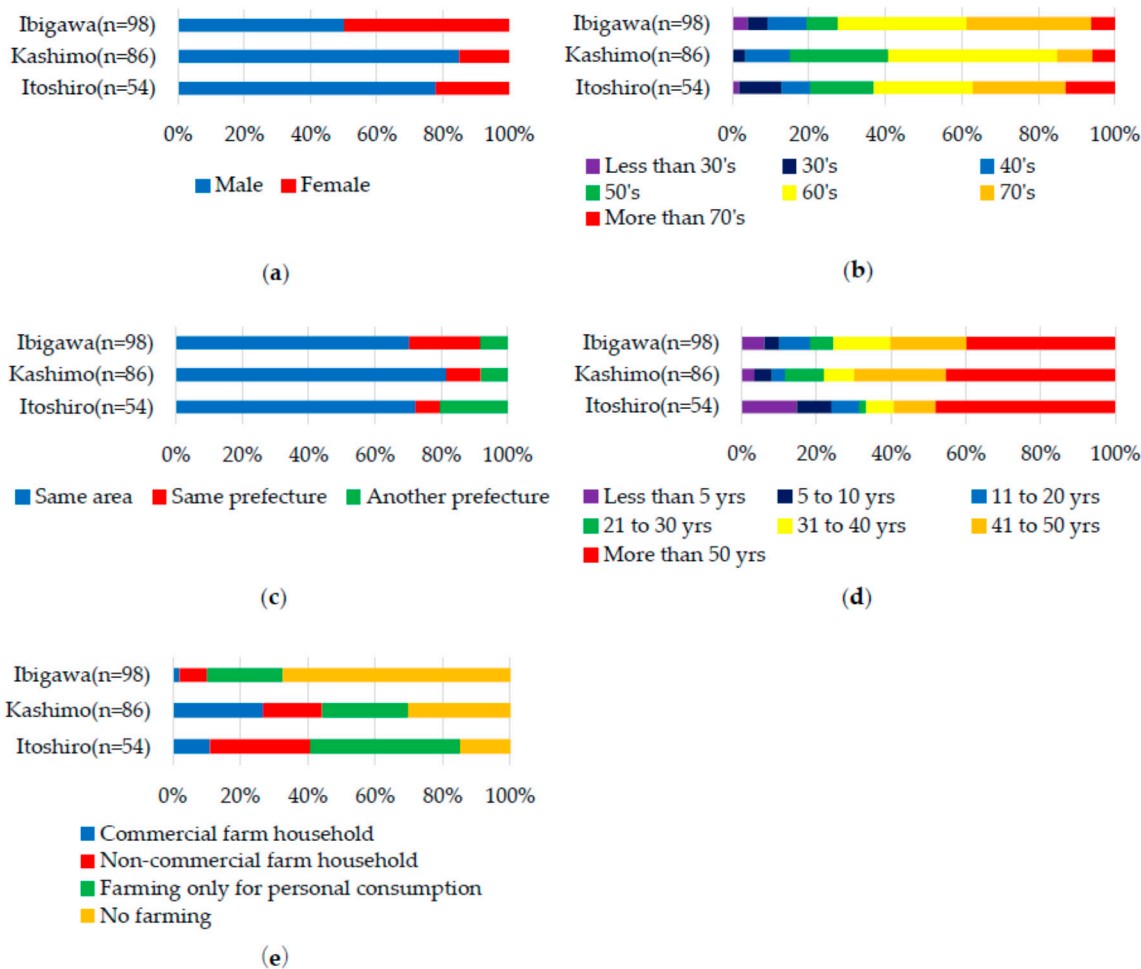

**Figure 1.** Profile of respondents: (**a**) Sex; (**b**) Age; (**c**) Hometown; (**d**) Years lived in the area; (**e**) Farming activity.

More than 70% of the respondents in Itoshiro were from the local community, and 20% were from another prefecture; this was the largest proportion of people from another prefecture in the three areas. More than 80% of the respondents in Kashimo were from the local community; the largest proportion in that category. The proportion from the local community in Ibigawa was similar to that in Itoshiro, but the percentage of residents from another community in the same prefecture was largest, at 21%.

Almost half of the respondents in Itoshiro (47%) had lived there for more than 50 years, and eight respondents had lived there for less than 5 years. In Kashimo, most respondents had lived there long-term, and 88% had lived there for more than 30 years. The distribution in Ibigawa was similar to that in Kashimo, but the proportions of respondents who had lived in Ibigawa for less than 5 years, 10–20 years or 30–40 years were twofold greater than those in Kashimo.

In Itoshiro, 85% of respondents engaged somewhat in farming, but the maximum percentage of commercial farming households was 11%. The proportion of commercial farm households was the largest (27%) in Kashimo. Both commercial and non-commercial farming households were relatively rare in Ibigawa, and two-thirds of the respondents there did not engage in farming at all.

3.2.3. Environmental Awareness

The responses regarding environmental awareness of such topics as domestic energy issues, local environmental issues and regional promotions are shown in Figure 2. The most frequent answer to the question regarding domestic energy issues was "interested but not active" in all three areas; the rates

were 57% (Itoshiro), 64% (Kashimo) and 61% (Ibigawa). The proportion of "not interested" residents was highest in Ibigawa (21%).

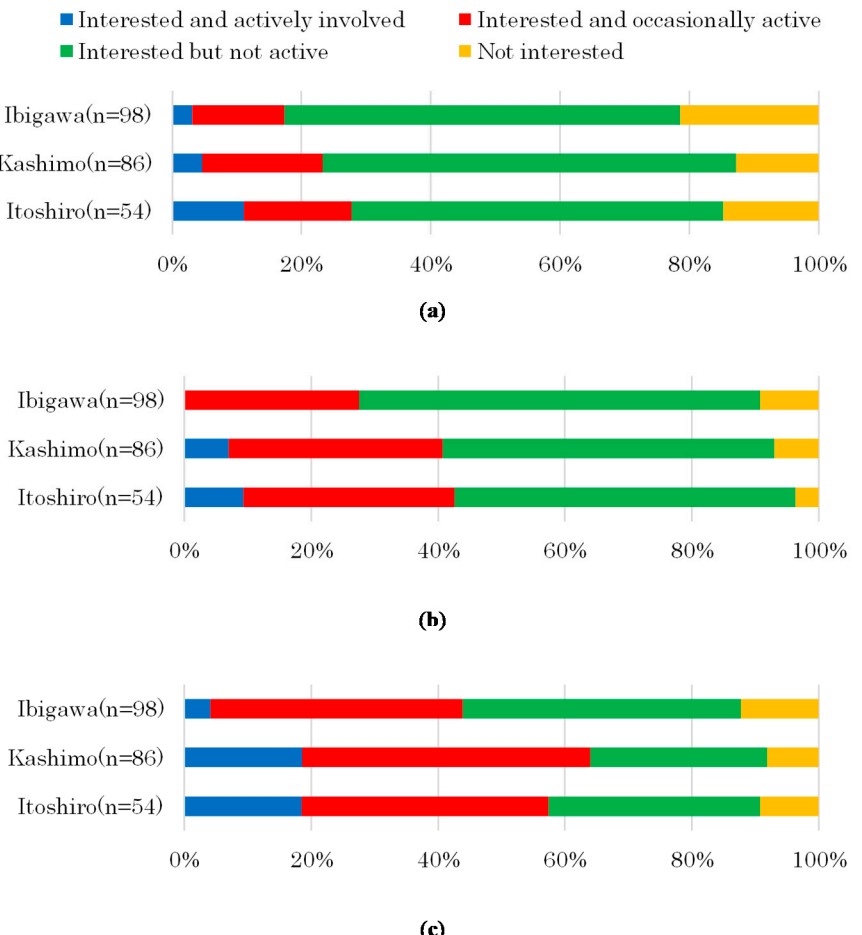

**Figure 2.** Responses to questions on environmental awareness in each area: (**a**) Domestic energy issues; (**b**) Local environmental issues; (**c**) Regional promotion.

The most frequent answer to questions regarding local environmental issues was also "interested but not active" in all three areas; 54% (Itoshiro), 52% (Kashimo) and 63% (Ibigawa). The numbers of respondents who selected "interested and actively involved" or "interested and occasionally active" were higher in Itoshiro and Kashimo than in Ibigawa. In particular, the rates of people responding "interested and actively involved" were 9% (Itoshiro) and 7% (Kashimo), whereas no respondents chose this answer in Ibigawa.

The responses "interested and actively involved" and "interested and occasionally active" to the question about regional promotion were higher in all areas compared to these responses to the questions about domestic energy and local environmental issues.

### 3.2.4. Participation in and Recognition of Projects

The responses regarding general knowledge about small hydropower, and recognition of and participation in local projects, are shown in Figure 3. Almost half of the respondents knew about small hydropower; only 12% did not know about it. In Itoshiro and Kashimo, approximately 60% of the respondents were quite familiar with small hydropower, whereas one-third of the respondents in Ibigawa did not know about it.

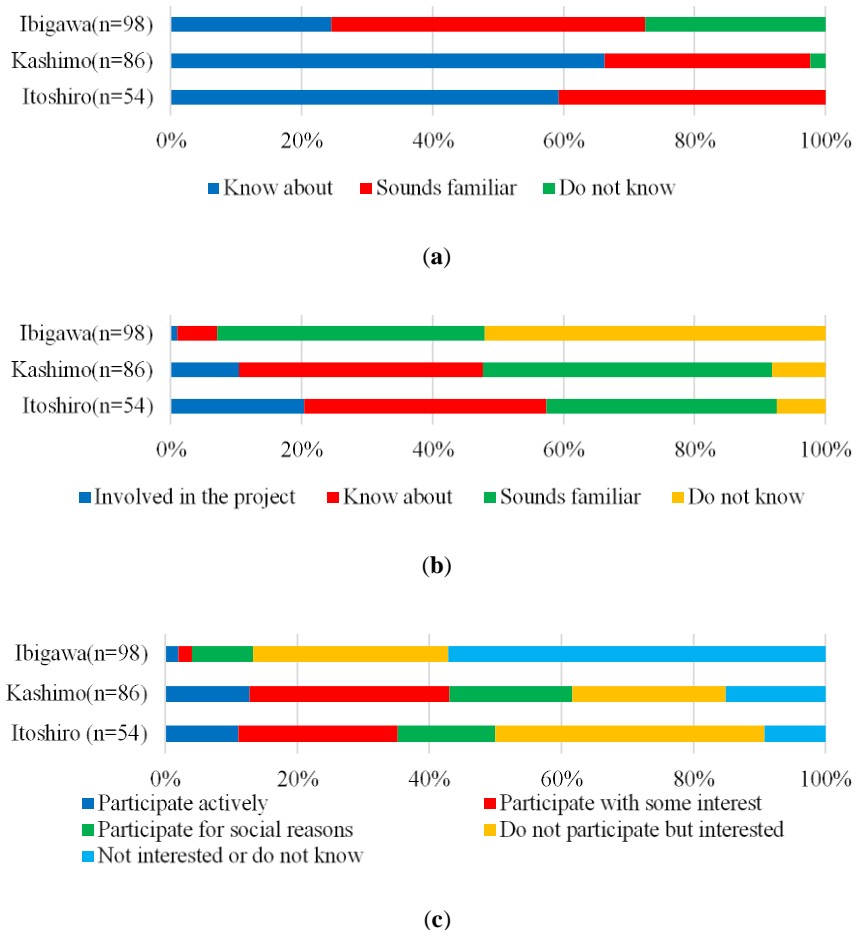

**Figure 3.** Responses to general knowledge about small hydropower and recognition and participation in the local project: (**a**) General knowledge on small hydropower; (**b**) Recognition of the local project; (**c**) Participation in the local project.

Local projects were recognized by more than 90% of the respondents in Itoshiro and Kashimo, whereas more than half of those in Ibigawa did not know about local projects. The rate of "sounds familiar" responses was approximately 40% in all three areas.

The rate of participation (participate actively + participate with some interest + participate for social reasons) was highest in Kashimo, at 62%, followed by Itoshiro at 50%. It was lowest in Ibigawa at 13%. Although the rates of recognition and participation were low, the response "do not participate but interested" was chosen by fewer than 30% of respondents in Ibigawa.

Cross tabulations of general knowledge, regarding small hydropower projects, and recognition of and participation in local projects, are shown in Tables 4 and 5. Almost all of the 169 respondents who somehow recognized the local project ("involved in the project (n = 19 + 2)"+"know about it (n = 50 + 8)" + "sounds familiar (n = 37 + 53 + 7)") had some knowledge of small hydropower, while only 7 respondents merely "recognized" the local project (see Table 4). All respondents who participated in the local project ("participate actively (n = 8 + 10 + 1)"+"participate with some interest (n = 11 + 23 + 7)"+"participate for social reasons (n = 2 + 9 + 22)") recognized it, regardless of their attitude (see Table 5).

**Table 4.** Cross tabulation of general knowledge about small hydropower and recognition of local projects.

| | | General Knowledge on the Small Hydropower | | |
| --- | --- | --- | --- | --- |
| | | **Know about** | **Sounds Familiar** | **Do Not Know** |
| Recognition of the local project | Involved in the project | 19 | 2 | 0 |
| | Know about | 50 | 8 | 0 |
| | Sounds familiar | 37 | 53 | 7 |
| | Do not know | 7 | 33 | 22 |

**Table 5.** Cross tabulation of recognition of and participation in the local project.

| | | Recognition | | | |
| --- | --- | --- | --- | --- | --- |
| | | **Involved in the Project** | **Know about** | **Sounds Familiar** | **Do Not Know** |
| Participation | Participate actively | 8 | 10 | 1 | 0 |
| | Participate with some interest | 11 | 23 | 7 | 0 |
| | Participate for social reasons | 2 | 9 | 22 | 0 |
| | Do not participate but interested | 0 | 16 | 51 | 4 |
| | Not interested or do not know | 0 | 0 | 16 | 58 |

From these results, the respondents were categorized into four groups: participation (n = 93: 39%), recognition (n = 76: 32%), knowledge (n = 40: 17%) and control (n = 29: 12%). The groups were considered in this order; for example, the "participation" respondents recognized the local project and knew about small hydropower, while the "recognition" respondents did not participate in the local project but knew about small hydropower. The seven respondents who answered "sounds familiar" regarding the local project and "do not know" about small hydropower were considered controls, because they neither participated, nor expressed interest, in the local project.

### 3.2.5. Effects of Respondents' Environmental Awareness

The responses to environmental awareness, in each category of concern, with regard to small hydropower projects are shown in Figure 4. The respondents exhibited a high degree of environmental awareness in all three respects surveyed, in the order of participation, recognition, knowledge and control.

A one-sided test was conducted regarding environmental awareness; the results are shown in Table 6. The degrees of environmental awareness were significantly higher in the participation category than in the recognition category; 95% for domestic energy issues and 99% for local environmental issues and regional promotion. With respect to domestic energy issues and regional promotion, the degrees of awareness were significantly higher in the recognition category than in the knowledge category, at the 90% level.

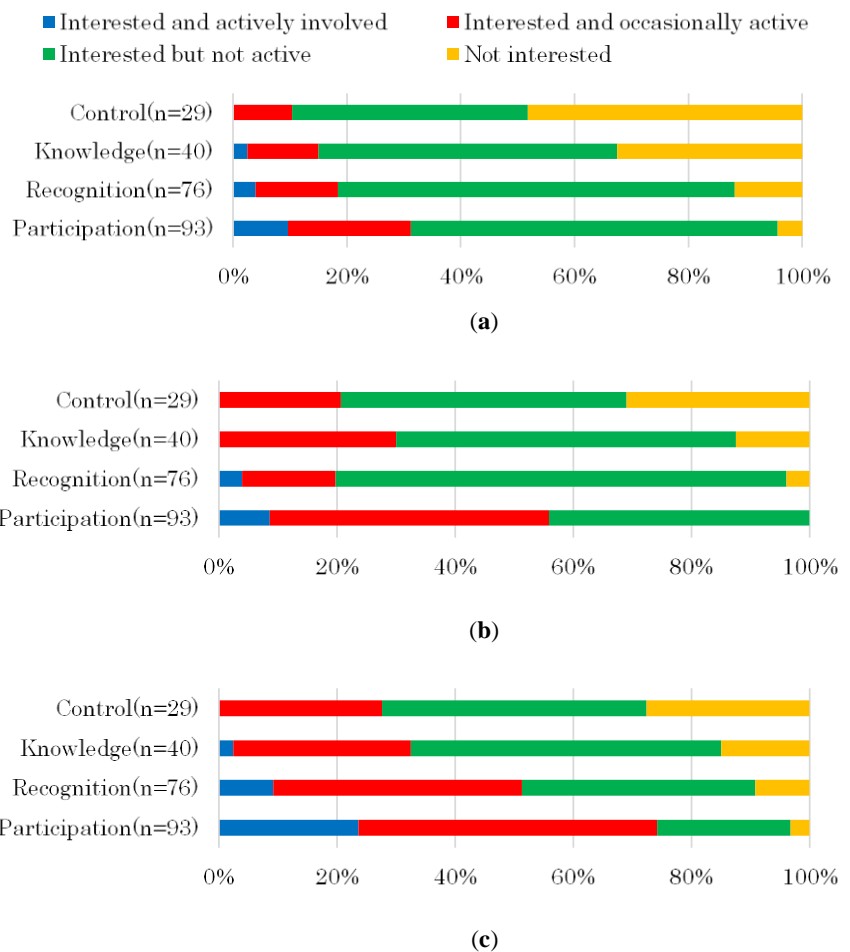

**Figure 4.** Environmental awareness in each category of involvement with small hydropower projects: (**a**) Domestic energy issues; (**b**) Local environmental issues; (**c**) Regional promotion.

**Table 6.** *p*-values of the one-sided U-test of the responses to the environmental awareness according to the involvement categories.

|  | Participation–Recognition | Recognition–Knowledge | Knowledge–Control |
|---|---|---|---|
| Domestic energy issues | 0.008 ** | 0.019 * | 0.104 |
| Local environment issues | Less than 0.001 *** | 0.579 | 0.065 |
| Regional promotion | Less than 0.001 *** | 0.020 * | 0.168 |

Significance was tested by adjusted criteria for triple comparison by Bonferroni correction. Symbols of *, ** and *** indicate 90%, 95% and 99% significance, respectively.

## 4. Discussion and Conclusions

We found that respondents exhibited high degrees of environmental awareness in participation, recognition, knowledge, and control across all three topics surveyed (domestic energy issues, local environmental issues and regional promotion (Figure 4)). This suggests that greater involvement consistently increases environmental awareness [22,23,33]. For example, Amponsah et al. [23] reported that participation in the project increased awareness, compared to simple recognition, from prior explanations or daily public relations. Furthermore, we found that recognition of local projects increased environmental awareness, compared to general knowledge of small hydropower. This implies that the prior explanation and/or public relations of local projects should work towards environmental education, as well as the original purposes of those activities. This finding expands the current knowledge regarding the relationship between participation and environmental awareness.



In addition, the degrees of awareness were significantly higher in the recognition category than in the knowledge category for domestic energy issues and regional promotion. The degrees of environmental awareness were significantly higher in the participation category than in the recognition category for the three aspects studied: domestic energy issues, local environmental issues and regional promotion (Table 6). The degrees of awareness were significantly higher in the recognition category than in the knowledge category for domestic energy issues and regional promotion. Because small hydropower projects in Itoshiro, Kashimo and Ibigawa promoted functions other than energy production (Table 2), prior explanations and daily public relations presumably raised awareness of domestic energy issues and supported regional promotion. Furthermore, participation in projects raises the awareness of local environmental issues, because common problems affecting small hydropower plants and related facilities include trash management in the irrigation canal (e.g., waste cans and fallen leaves or branches).

To the best of our knowledge, this is the first study to investigate the effects of participation in small hydro development related to existing irrigation systems on environmental awareness. Notably, the causality between involvement and environmental awareness is not clear from the results of this study; however, inclusive relationships between respondent involvement and awareness were revealed. This suggests that the trigger for resident involvement is a key factor in developing small hydro in existing irrigation systems because knowledge and recognition is a first step towards deeper involvement and participation. Thus, intentional acceleration of the spiral feedback of participation—knowledge and experience—could help ensure that small hydropower plants are used as multifunctional regional resources, and thus involve more non-farmers. To this end, collaboration between project management bodies and other organizations in environmental education can act as a trigger for resident involvement.

A limitation of this study is that the respondents were mainly older and male (Figure 1), and that the effects of participation on awareness were not validated for all the residents. When we expand existing irrigation systems to multifunctional regional resources by installing small hydro facilities, the involvement of new stakeholders is important, and the validation (for all ages and genders) of our findings on the inclusive relationships between respondent involvement and awareness is recommended for future research.

**Author Contributions:** Conceptualization, K.N.; methodology, K.N., K.M., K.I., M.S.; formal analysis, K.N.; investigation, K.M.; resources, M.S.; writing—original draft preparation, K.N., K.M.; writing—review and editing, K.N.; visualization, K.N.; supervision, M.S.; project administration, K.N., M.S.; All authors have read and agreed to the published version of the manuscript.

**Funding:** This research was performed by the Environment Research and Technology Development Fund (JPMEERF20202004) of the Environmental Restoration and Conservation Agency of Japan.

**Acknowledgments:** The authors would like to thank the interviewees of the management organizations, the NPO of Yasuragi-no-sato Itoshiro, the Ogo irrigation association and all the respondents to the questionnaire survey.

**Conflicts of Interest:** The authors declare no conflict of interest.

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
