# Peer review of "Effect of Residents’ Involvement with Small Hydropower Projects on Environmental Awareness"

_sustainability, doi:10.3390/su12155994_

Round 1

Reviewer 1 Report

With the new introduction, this article has become far more interesting and relevant to the reader

Author Response

Thank you very much.

Reviewer 2 Report

The paper has been improved significantly and it is obvious that the authors have dedicated time to revising their work. However, there is still some room for improvement and hopefully my comments will be helpful.

  1. The authors should redefine the objectives their study tries to meet.
  2. The sentence in lines 33-34 is a bit vague and needs to be rephrased so that the relationship between renewables and local availability is explained.
  3. Towards the end of the Introduction, the literature gap that this study tries to fill needs to be highlighted.
  4. In the last paragraph of the Introduction, the study objectives must be stated loud and clear.
  5. What does the acronym NPO stand for? And what exactly is it?
  6. In the Introduction (rouphly line 79), the authors could provide some information on the current electricity system in Japan. Which fossil fuels and renewables consist the Japanese energy mix over the last year? What is the percentage of each energy source in the mix? What is the role of hydropower in electricity?
  7. In line 115, the authors should perhaps replace "Population aging" with "Aging population".
  8. On line 216, can the authors explain what they mean by stating that "the survey was conducted indirectly"?
  9. The Discussion and Conclusions section needs to be revised and enriched as it is short and weak now. In essence, the authors do not discuss their findings. The findings need to be contextualized and interpreted but not just repeated. The authors do not attribute the findings to anything nor they compare them to relevant studies. In addition, they do not state what might have affected respondents' attitudes. It is also important to state clearly whether the study objectives have been met and how these insights could promote the development of hydropower or renewable energy in general. Based on specific findings and the discussion the authors could then reach meaningful conclusions. Finally, in view of their findings the authors could make some recommendations for future research. 

Author Response

Response to Reviewer 2 Comments

Point 1: The authors should redefine the objectives their study tries to meet. 

Response 1: Many thanks for the patient comment (relating Point 3 &4). We have moved a paragraph from the previous Discussion and conclusions to clarify the literature gap and the objectives of this study (L.114-122)

Point 2: The sentence in lines 33-34 is a bit vague and needs to be rephrased so that the relationship between renewables and local availability is explained.

Response 2: Thank you for the comment. The following sentence intends to explain the relationship between renewables and local availability. In order to clarify it, we have improved it as follows (L.34-38):

Point 3: Towards the end of the Introduction, the literature gap that this study tries to fill needs to be highlighted.

Response 3: Please refer to the Response 1.

Point 4: In the last paragraph of the Introduction, the study objectives must be stated loud and clear.

Response 4: Please refer to the Response 1.

Point 5: What does the acronym NPO stand for? And what exactly is it?

Response 5: Thank you. We have added the definition of NPO as non-profitable organization. (L.131)

Point 6: In the Introduction (rouphly line 79), the authors could provide some information on the current electricity system in Japan. Which fossil fuels and renewables consist the Japanese energy mix over the last year? What is the percentage of each energy source in the mix?

Response 6: Thank you for the comment. Accordingly, we have added the explanation of the current situation of Japanese energy mix as follows (L.73-78):

From

“In Japan, the development of renewable energies is strongly promoted by the government, especially following the 2011 nuclear accident [23,24]. The share of renewable energy contributing to electricity generation increased from 9% in 2010 to 17% in 2018, with the majority coming from solar and biomass (87% and 12% of the gains, respectively) [25].”

To

“In Japan, the development of renewable energies is strongly promoted by the government, especially following the 2011 nuclear accident [23,24]. In 2018, fossil fuels, renewables and nuclear account for 77% (natural gas: 38%, coal: 32%, coal oil: 7%), 17% (hydro: 8%, solar: 6%, biomass: 2%, wind: 1%, geothermal: less than 1%) and 6%, respectively; the share of renewable energy contributing to electricity generation increased by 8% from 2010 to 2018, with the majority coming from solar and biomass (87% and 12% of the gains, respectively) [25].”

Point 7: In line 115, the authors should perhaps replace "Population aging" with "Aging population".

Response 7: Thank you. In accordance with the Reviewer’s comment, we have replaced “Population aging and decline” into “Aging and falling population”. (L.129)

Point 8: On line 216, can the authors explain what they mean by stating that "the survey was conducted indirectly"?

Response 8: Thank you. We intend by stating that “the survey was conducted indirectly” to contrast the survey in the Itoshiro and Kashimo with that in the Ibigawa area in a direct visit (face to face). (L.229-233)

Point 9: The Discussion and Conclusions section needs to be revised and enriched as it is short and weak now. In essence, the authors do not discuss their findings. The findings need to be contextualized and interpreted but not just repeated. The authors do not attribute the findings to anything nor they compare them to relevant studies. In addition, they do not state what might have affected respondents' attitudes. It is also important to state clearly whether the study objectives have been met and how these insights could promote the development of hydropower or renewable energy in general. Based on specific findings and the discussion the authors could then reach meaningful conclusions. Finally, in view of their findings the authors could make some recommendations for future research.

Response 9: Many thanks for the patient comment. In accordance with the Reviewers suggestions, we have added some sentences in Discussion and conclusions (L.372-373, L.399-404)

Reviewer 3 Report

The paper discusses a survey about the acceptance and population point of view about small hydropower, based on real installations.

I have a list of  comments that have to be taken into account. See the attached file for specific comments.

The English is good and clear.

1) specify the country of the author affiliation.

2) One key example of small hydro in irrigation canals is the use of gravity water wheels, as new installations or rehabilitation of old water mills. See for example https://www.sciencedirect.com/science/article/pii/S1364032118306178

3) line 90: results of this reference should be briefly described.

4) define NPO.

5) line 118: use capital letters for the plant/city first letter name.

6) chapter 2.1, for each hydro plant: specify the plant characteristics, or refer to Table 1.

7) define LID.

8) Table 1: specify the turbine type if possible.

9) add more references and a brief description about the Mann–Whitney U-test and the Bonferroni correction.

10) Table 2: "Coverage of prior explanation" to be better explained.

11) define the "response rate".

12) Fig 1b and 1d: define the legend (is it the age?).

13) Figs. 1-2-3-4: the quality is very low, especially the text. Such low quality is not acceptable for a scientific paper.

14) Lines 340-348: this should be written in the Introduction in order to identify the novelty and originality of this work.

15) more discussion about the effect of age and gender should be presented.

Author Response

Response to Reviewer 3 Comments

Point 1: specify the country of the author affiliation.

Response 1: Thank you. We have added the countries of the authors’ affiliation (L.5-8)

Point 2: One key example of small hydro in irrigation canals is the use of gravity water wheels, as new installations or rehabilitation of old water mills. See for example https://www.sciencedirect.com/science/article/pii/S1364032118306178

Response 2: Thank you for introducing a significant paper. We have added the explanation in Introduction as follows (L.47-49):

“Quaranta and Revelli [11] reviewed on gravity water wheels which are efficient and cost-effective micro hydropower converters.”

Point 3: line 90: results of this reference should be briefly described.

Response 3: Thank you. The objective of Hirose et al. is to assess the non-farmers’ preferences and the essence is explained in the next sentence. (L. 95-98)

“Hirse et al. [31] evaluated the preferences of non-farmers for water wheels used in an irrigation canals in Okayama Prefecture, Japan. They extracted three factors that appeared to influence residents' preferences: regionality, functionality, and environmental friendliness.”

Point 4: define NPO.

Response 4: Thank you. We have added the definition of NPO as non-profitable organization. (L.131)

Point 5: line 118: use capital letters for the plant/city first letter name.

Response 5: Thank you. We have revised the expression as Itoshiro-Banba-Seiryu and Kashimo-Seiryu hydropower plant (L.133 and 139).

Point 6: chapter 2.1, for each hydro plant: specify the plant characteristics, or refer to Table 1.

Response 6: In accordance with the Reviewer’s comment, we have added the reference to Table 1 for each hydro plant (L.133, 139 and 144).

Point 7: define LID.

Response 7: Thank you. LID is defined in L.85-89.

Point 8: Table 1: specify the turbine type if possible.

Response 8: Thank you. In accordance with the Reviewer’s comment, we have added a row of “Turbine type” in Table 1.

Point 9: add more references and a brief description about the Mann–Whitney U-test and the Bonferroni correction.

Response 9: Thank you. In accordance with the Reviewer’s comment, we have added a reference about Bonferroni corrected Mann-Whitney U-test (L.180)

Kotrshal K., Hirchenhauser K., Mostl E. The relationship between social stress and dominance is seasonal in graylag geese. Anim Behav 1998, 55, 171-176.

Point 10: Table 2: "Coverage of prior explanation" to be better explained.

Response 10: Thank you. In accordance with the Reviewer’s comment, we have revised the name of the Item from “Coverage of prior explanation” to “Coverage of prior explanation of the project.” (Table 2)

Point 11: define the "response rate".

Response 11: Thank you. In accordance with the Reviewer’s comment, we have added the definition of the response rate as “response rates (=valid responses/distribution*100)”  (L.227)

Point 12: Fig 1b and 1d: define the legend (is it the age?). 

Response 12: Thank you for the comment. We have revised the legend in Fig 1b and 1d to clarify the age and years respectively.

Point 13: Figs. 1-2-3-4: the quality is very low, especially the text. Such low quality is not acceptable for a scientific paper.

Response 13: We are sorry for the low quality of the figures. The quality of the figures seems to deteriorate in the process of the format conversion from WORD to PDF. This time, we submit the figures in PDF to keep the quality.

Point 14: Lines 340-348: this should be written in the Introduction in order to identify the novelty and originality of this work..

Response 14: Thank you for the comment. In accordance with the Reviewer’s comment, we have moved the fist paragraph to the end of Introduction so as to identify the novelty and originality of this paper (L.114-122).

Point 15: more discussion about the effect of age and gender should be presented.

Response 15: Many thanks for the patient comment. We recognise the bias of respondents in age and gender as a limitation of this study and have added a paragraph at the end of Discussion and Conclusion in order to clarify it (L.399-404).

Round 2

Reviewer 3 Report

The paper is improved, but I suggest the following minor revisions:

- Response 2

Quaranta and Revelli [11] not present in the references.

- Response 3

Briefly discuss the results in few lines.

- Response 8

"Spiral water wheel" should be better defined. Do you mean breastshot water wheels? See the suggested reference (Quaranta and Revelli, 2018, or Muller and Kauppert, 2004 on water wheels)

- Response 12

The difference between years and age should be better defined in the caption.

- Delete the empty lines 347-348

My final feeling is that the scientific value and novelty of the paper is not high enough to be published as a scientific paper, since no novel results have been added. The manuscript should be published as a different article type.

Author Response

Response to Reviewer 3 Comments

Point 2: One key example of small hydro in irrigation canals is the use of gravity water wheels, as new installations or rehabilitation of old water mills. See for example https://www.sciencedirect.com/science/article/pii/S1364032118306178

Response 2: Thank you for introducing a significant paper. We have added the explanation in Introduction as follows (L.47-49):

“Quaranta and Revelli [11] reviewed on gravity water wheels which are efficient and cost-effective micro hydropower converters.”

Point 2-2: Quaranta and Revelli [11] not present in the references.

Response 2: I am sorry for the point. We have added Quaranta and Revelli [11] in the references and adjusted the related reference numbers.

Point 3: line 90: results of this reference should be briefly described.

Response 3: Thank you. The objective of Hirose et al. is to assess the non-farmers’ preferences and the essence is explained in the next sentence. (L. 95-98)

“Hirose et al. [31] evaluated the preferences of non-farmers for water wheels used in an irrigation canals in Okayama Prefecture, Japan. They extracted three factors that appeared to influence residents' preferences: regionality, functionality, and environmental friendliness.”

Point 3-2: Briefly discuss the results in few lines.

Response 3-2: Thank you. We have added the discussion of results of Hirose et al. [31] as follows;

“Hirose et al. [31] evaluated the preferences of non-farmers for water wheels used in an irrigation canals in Okayama Prefecture, Japan. They extracted three factors that appeared to influence residents' preferences: regionality, functionality, and environmental friendliness. They suggested that the practical use of the water wheels for the irrigation purpose was important in terms of functionality, even though the function did not benefit the non-farmers.”

Point 8: Table 1: specify the turbine type if possible.

Response 8: Thank you. In accordance with the Reviewer’s comment, we have added a row of “Turbine type” in Table 1.

Point 8-2: "Spiral water wheel" should be better defined. Do you mean breastshot water wheels? See the suggested reference (Quaranta and Revelli, 2018, or Muller and Kauppert, 2004 on water wheels)

Response 8-2: Thank you, but we did not the same water wheel in suggested references. Here, the sketch below is  the spiral water wheel (Takimoto, 2010).

Point 12: Fig 1b and 1d: define the legend (is it the age?).

Response 12: Thank you for the comment. We have revised the legend in Fig 1b and 1d to clarify the age and years respectively.

Point 12-1: The difference between years and age should be better defined in the caption.

Response 12-1: Thank you. We have revised the caption as “Figure 1 Profile of respondents: (a) Sex; (b) Age (Year old); (c) Hometown; (d) Years to live in the area; (e) Farming activity.”

- Delete the empty lines 347-348

Thank you. We have deleted the empty lines.

This manuscript is a resubmission of an earlier submission. The following is a list of the peer review reports and author responses from that submission.

Round 1

Reviewer 1 Report

Please explain the need for a Bonferroni correction in this case.

Reviewer 2 Report

  • Overall it is an interesting paper and presents an interesting topic. However, there are major issues that need to be addressed before the paper is considered for publication again.
  • Although the Introduction of the paper is quite informative, it does not present the context of the study nor it presents the topic adequately. In addition, the authors do not mention the literature gap they try to fill with their research and do not state the aim of the study. Hence, the Introduction needs to be improved significantly. In addition, in this section the authors could also present the energy situation in Japan (shares of renewable and non-renewable energy in the mix) and mention briefly the energy policy on renewable energy. Moreover, they could explain why social research on renewable energy is important.
  • In the Introduction, the authors also refer to some research findings of relevant studies. However, the relevant literature works should be presented in a separate section. That is, there should be a separate section titled “Literature review” or “Theoretical background” in which the relevant literature works should be presented so that the authors can compare this information to their own findings in the Discussion. This section (where the relevant literature is presented) should include a sufficient number of references and be at least two pages long.
  • Subsection 2.1 should be titled “Study area” instead of “Target area”.
  • In line 101, the word “profiles” should be replaced with “demographic characteristics”.
  • In subsection 2.3 where the authors describe the content of the questionnaire, they refer to “three-step ordered responses”, but do not state if this is the Likert scale. Hence, the should explain more the response scale used in the questionnaire.
  • Which sampling method was used to collect the data in Itoshiro and Kashimo? Are all three samples representative? Which formulae were used to estimate the sample sizes for each study area? This information must be stated loud and clear in the Materials and Methods section. All information on the methodology followed to conduct the study must be precise and clear.
  • In line 139, the authors must explain what the Environmental Policy Division is.
  • The results of the interviews are written vaguely, and the important points are not stressed enough.
  • The authors should ensure that they use the correct terms in Table 3.
  • In the Discussion, the authors in essence present research findings and even include tables. The findings must be presented and described in detail only in the Results section. In the Discussion, the authors should contextualize and interpret their findings and compare them to the results of the relevant literature.
  • The Conclusions section is in essence a very short discussion with a few concluding remarks. In their Conclusions, the authors should communicate clear the conclusions which can be drawn based on specific study findings.
  • In terms of English, there are errors in grammar and syntax while many sentences are too short and fail to communicate the meaning.

Reviewer 3 Report

The manuscript "Effect of residents’ involvement with small hydropower projects on environmental awareness" brings a interesting assessment of riverine communities nearby small hydropower plants in Japan.

While the study subjet is interesting, the manuscript has some limitation on study desing, study area description and statistical analysis that must be addressed. Therefore I recoment to reject and resubimit.

Specifics comment:

Abstract

The abstract must bring important information about the study methods and results. Foi instance: How many people answered the survey? How is the most important conclusion of the study?

Keywords

The keywors should not be repeated in the title, like small hydropower and environmental awareness.

Introduction

The introduction it is poor. The introduction need a deep review to bring a overview on the effect of hydropower plants around the world incluing environmental e socical impacts. There are a large amount of published paper about the hydropower developmento in Sout America, Africa, Europe...

The Sustainability is a Journal of global audience and the introduction must be support by global evidence that support your assertations.

Besides that, the introduction dos not bring basic elements of a paper like: hypotesis, premisse or prediction.

Material and methods

The manuscript needs a deep study area characterization. For example, the river name, the size of riverine communities. These kind of information is essential for manuscrit improvement.
Other importants poins is, How was choiced the residences to apply the quiz? It was randomized?

How many people were interviewed in each house? Even it seems insignificat, if the interview was done with more than one people of a residence the results could be biased. People from same family nucleus tend to have the same perception on politic, economic, nature...

These are very important information that must be clear in the paper for facilite the review process and the reader comphreension of paper.

Another problem is the statisctical analysis. I think that the mann-whitney is not appropriated for proportion variables. Maybe a proportion test like binomial z, chi-squere or contingency table can solve these problem...

Discussion

The discussion needs a deep review. The discussion isn't ground by a robust scientific information. Despite of, the discussion seem a continuation of results, bringing more data, figures and tables.

Conclusion

The manuscript conclusion begins with a short introduction. However, the conclusion should be direct and try to describe the most important finds of the manuscript.